Pseudorhabdosynochus sulamericanus (Monogenea, Diplectanidae), a parasite of deep-sea groupers (Serranidae) occurs transatlantically on three congeneric hosts (Hyporthodus spp.), one from the Mediterranean Sea and two from the western Atlantic

Chaabane Amira amirachaabene@hotmail.fr 1
Justine Jean-Lou 2
Gey Delphine 3
Bakenhaster Micah D. 4
Neifar Lassad 1
1 Faculté des Sciences de Sfax, University of Sfax , Sfax , Tunisia
2 ISYEB, Institut de Systématique, Évolution, Biodiversité, Muséum National d’Histoire Naturelle, Sorbonne Universités , Paris , France
3 UMS 2700 Service de Systématique Moléculaire, Muséum National d’Histoire Naturelle, Sorbonne Universités , Paris , France
4 Fish and Wildlife Research Institute, Florida Fish and Wildlife Conservation Commission , St. Petersburg , FL , USA
Riutort Marta
Electronic publication date: 2016 Aug 16
Publication date: 2016
Volume: 4
Electronic Location ID: e2233
Received 2016 Apr 12; Accepted 2016 Jun 17
Copyright: ©2016 Chaabane et al.
Copyright year: 2016
Copyright holder: Chaabane et al.
License: This is an open access article distributed under the terms of the Creative Commons Attribution License, which permits unrestricted use, distribution, reproduction and adaptation in any medium and for any purpose provided that it is properly attributed. For attribution, the original author(s), title, publication source (PeerJ) and either DOI or URL of the article must be cited.
License URL: https://creativecommons.org/licenses/by/4.0/

Keywords: Monogenea, Grouper, Mediterranean Sea, Geographic distribution, Barcoding, Deep-sea fish, Morphology, Fish parasites

Funding: BIOPARMED- ENVI-MED MNHN ATM Barcode State of Florida saltwater recreational fishing license revenues US Department of the Interior US Fish and Wildlife Service Federal Sportfish Restoration F-72, F-123 Travel expenses of AC were funded by the program BIOPARMED- ENVI-MED (http://www.mistrals-home.org/spip.php?rubrique82). Molecular work was funded by MNHN ATM Barcode (www.mnhn.fr). MB was funded by State of Florida saltwater recreational fishing license revenues (http://myfwc.com/license/recreational/saltwater-fishing/) and the US Department of the Interior US Fish and Wildlife Service Federal Sportfish Restoration Grants, F-72 and F-123 (http://wsfrprograms.fws.gov/). The funders had no role in study design, data collection and analysis, decision to publish, or preparation of the manuscript.

==============================
Little is known of the diversity of the monogenean parasites infesting deep-sea groupers, and there is even less information available about their geographic distributions within the ranges of their hosts. To improve our understanding of these host-parasite relationships we conducted parasitological evaluations of the deep-water Haifa grouper Hyporthodus haifensis from the southern Mediterranean off Tunisia and Libya. We collected more than one species of diplectanid monogeneans from this host, but among these only one dominant species was abundant. This proved to be morphologically very similar to Pseudorhabdosynochus sulamericanus Santos, Buchmann & Gibson, 2000, a species originally described from the congeneric host H. niveatus off Brazil and also recorded from H. niveatus and H. nigritus off Florida. Here, we conducted a morphological comparison between newly collected specimens and those previously deposited in museum collections by other authors. Further, we used COI barcoding to ascertain the specific identity of the three host species to better elucidate the circumstances that might explain the unexpectedly broad distribution of P. sulamericanus. We assigned our specimens from H. haifensis to P. sulamericanus primarily on the basis of morphological characteristics of the sclerotized vagina. We also noted morphological characteristics of eastern and western Atlantic specimens that are not clearly described or not given in previous descriptions and so prepared a redescription of the species. We confirmed, by COI barcoding, that no sister-species relationships were evident among the three hosts of P. sulamericanus. Our observation that P. sulamericanus infects unrelated host species with putatively allopatric distributions was unexpected given the very limited dispersive capabilities and the high degree of host specificity common to members of Pseudorhabdosynochus. This transatlantic distribution raises questions with regard to phylogeography and assumptions about the allopatry of Atlantic grouper species from the Americas and Afro-Eurasia. Here, we propose some hypothetical explanations for our findings.

Introduction

Groupers (Serranidae, Epinephelinae) are known to harbour rich parasitic fauna (Cribb et al., 2002; Justine et al., 2010), including an exceptionally high number of diplectanid monogenean species belonging to the genus Pseudorhabdosynochus Yamaguti, 1958 (Hinsinger & Justine, 2006; Justine, 2005a; Justine, 2005b; Justine, 2007a; Justine, 2007b; Justine, 2008c; Justine, 2010; Justine, Dupoux & Cribb, 2009; Justine et al., 2010; Justine & Sigura, 2007; Kritsky, Bakenhaster & Adams, 2015; Neifar & Euzet, 2007; Schoelinck & Justine, 2011; Zeng & Yang, 2007) and some species from other genera (Journo & Justine, 2006; Justine, 2007a; Justine, 2008a; Justine & Euzet, 2006; Justine & Henry, 2010; Sigura & Justine, 2008). Most grouper species live in tropical seas, particularly in coral reefs, and are thus shallow-water species. In these coral reef groupers, extremely high biodiversity of monogeneans has been reported and most monogenean parasites are strictly specific, being found in only one species of grouper (Justine, 2007a; Justine et al., 2010; Justine & Sigura, 2007). Some groupers, however, are deep-sea fish (Heemstra & Randall, 1993) and the parasites of these are poorly known. Recently, it was shown that two groupers from off New Caledonia shared the same species of Pseudorhabdosynochus. This was interpreted as the result of adaptation to deep sea by the monogenean because lower specificity helps transmission of the parasites in the deep-sea environment, where hosts are rarer than in coral seas (Schoelinck, Cruaud & Justine, 2012). This finding echoed previous results of lower biodiversity of monogeneans in deep-sea fish compared to surface fish (Rohde, 1988). However, depth gradients of diversity of parasites are not well known (Rohde, 2016).

In this paper, as part of a parasitological survey of groupers from the Mediterranean and the African Atlantic coast (Chaabane, Neifar & Justine, 2015; Moravec et al., 2016a; Moravec et al., 2016b; Neifar & Euzet, 2007), we studied the monogenean fauna of a deep-sea grouper from the Mediterranean Sea that had not previously been examined for parasites, the Haifa grouper, Hyporthodus haifensis (Ben Tuvia). We expected to recover previously unreported or undescribed species of Pseudorhabdosynochus from this fish, which is uncommon and poorly studied (Craig, Sadovy de Mitcheson & Heemstra, 2012). The single abundant species of Pseudorhabdosynochus found on H. haifensis revealed unexpected similarities to published descriptions of a species from the American coast of the Atlantic Ocean. We examined museum specimens, and conclude herein, from a comparative morphological study, that the grouper H. haifensis harbours P. sulamericanus Santos, Buchmann & Gibson, 2000 in the Mediterranean, a species reported only from the coast off Brazil and Florida (Kritsky, Bakenhaster & Adams, 2015; Santos, Buchmann & Gibson, 2000) on two other congeneric fish species, namely the snowy grouper H. niveatus (Valenciennes) and the Warsaw grouper H. nigritus (Holbrook). In contrast to these two species, H. haifensis is not known to occur in the western Atlantic (Froese & Pauly, 2016; Heemstra & Randall, 1993).

To interpret this unexpected finding, we ascertained that the three hosts H. haifensis, H. niveatus and H. nigritus were distinct species. To do this, we sequenced the COI gene, usually used for barcoding, of specimens of the two latter species from off the USA and of H. haifensis from the Mediterranean Sea off Tunisia and Libya. We tried to obtain specimens of the monogenean P. sulamericanus from the western Atlantic for a molecular analysis. Unfortunately, this was not possible and we could therefore not compare sequences with our COI sequences obtained from Mediterranean specimens. We also searched the monogenean literature for similar cases of transatlantic parasites. The present finding of the same monogenean species on different species of deep-sea fish on both sides of the Atlantic (South America vs Africa and the Mediterranean Sea) seems to be the first.

We propose hypotheses to explain this finding, some of which outline our very limited knowledge of the biology of deep-sea groupers.

Materials and Methods

Fish

Hyporthodus haifensis is a relatively rarely collected fish, and morphological differentiation from seemingly similar groupers is difficult (Craig, Sadovy de Mitcheson & Heemstra, 2012; Heemstra & Randall, 1993). Specimens of Hyporthodus haifensis were obtained from the fish markets of Sfax, Tunisia and Tripoli, Libya (Table 1). Field identifications of these specimens were made by the authors (AC & LN) with usual keys (Craig, Sadovy de Mitcheson & Heemstra, 2012; Heemstra & Randall, 1993; Louisy, 2015). One of our fish (Hh4; Table 1) was transported frozen from Tunisia to Paris where its identification was confirmed by thorough morphological analysis (Dr. B Séret, MNHN-IRD, pers. comm., 2015), and it was deposited as a voucher specimen in the ichthyological collection of the MNHN (registration number MNHN 2015-0242). For nine specimens of H. nigritus and eight specimens of H. niveatus (Table 2), tissue samples (fin clips) were collected from Gulf of Mexico commercial or recreational fisheries catches during routine fish population monitoring surveys conducted by the Florida Fish and Wildlife Conservation Commission (FWC); Ids in Table 2 correspond to collection numbers of FWC. Morphological identifications of Florida specimens were made by trained FWC fisheries biologists. Fish nomenclature follows FishBase (Froese & Pauly, 2016).

Table 1 Haifa grouper, Hyporthodus haifensis.

Fish examined, barcode sequences and diplectanid monogeneans collected.

Fish Id	Date	Locality	COI sequence	Fish state	Standard length (cm)	Pseudorhabdosynochus sulamericanus, number of specimens	Specimens of other, unidentified, Pseudorhabdosynochus species	
Hh1	27-04-2012	Tunisia	KT023566	Whole	55	71	1	
Hh2	01-06-2013	Libya	KT023567	Whole	70	90	3	
Hh3	03-06-2013	Libya	–	Gills	–	59	1	
Hh4	11-07-2014	Libya	KT023568	Wholea	76	123	0	
						Total: 343	Total: 5	
Notes.

a Fish specimen deposited in MNHN as MNHN 2015-0242.

Table 2 Snowy grouper H. niveatus and Warsaw grouper H. nigritus.

Origin of fish used for barcoding.

Id	Species	Locality	Collection date	GenBank	
Hnig_12Nov2015-01	Hyporthodus nigritus	Gulf of Mexico (GOM), off southern Florida, USA	11-11-2015	KU739508	
Hnig _CK133220	Hyporthodus nigritus	GOM, off central Florida, USA	12-12-2013	KU739507	
Hnig_037-01	Hyporthodus nigritus	GOM, off southern Florida, USA	24-07-2015	KU739504	
Hnig_076-01	Hyporthodus nigritus	GOM, off southern Florida, USA	unknown	KU739501	
Hnig_CK1303221	Hyporthodus nigritus	GOM, off central Florida, USA	12-12-2013	KU739502	
Hnig_CK1303222	Hyporthodus nigritus	GOM, off central Florida, USA	12-12-2013	KU739506	
Hnig_CK1303223	Hyporthodus nigritus	GOM, off central Florida, USA	12-12-2013	KU739509	
Hnig_13Nov2015-01	Hyporthodus nigritus	GOM, off Florida Keys, USA	13-11-2015	KU739505	
Hnig_30Oct2015-01	Hyporthodus nigritus	GOM, off southern Florida, USA	30-10-2015	KU739503	
Hniv_PE1400561	Hyporthodus niveatus	GOM, off Alabama, USA	20-02-2014	KU739511	
Hniv_079-01	Hyporthodus niveatus	GOM, off southern Florida, USA	XX-11-2015	KU739513	
Hniv_087-01	Hyporthodus niveatus	GOM, off southern Florida, USA	XX-11-2015	KU739517	
Hniv_097-01	Hyporthodus niveatus	GOM, off southern Florida, USA	XX-11-2015	KU739512	
Hniv_101-01	Hyporthodus niveatus	GOM, off southern Florida, USA	XX-11-2015	KU739514	
Hniv_103-01	Hyporthodus niveatus	GOM, off southern Florida, USA	XX-05-2012	KU739516	
Hniv_May2012-02	Hyporthodus niveatus	GOM, off northern Florida, USA	XX-05-2012	KU739510	
Hniv_May2012-03	Hyporthodus niveatus	GOM, off northern Florida, USA	XX-05-2012	KU739515	

Molecular barcoding of fish

We used the QIAamp DNA Mini Kit (Qiagen), per the manufacturer’s instructions, to perform DNA extraction. The 5′ region of the cytochrome oxidase I (COI) mitochondrial gene was amplified with the primers FishF1 (5′-TCAACCAACCACAAAGACATTGGCAC-3′) and FishR1 (5′-TAGACTTCTGGGTGGCCAAAGAATCA-3′) (Ward et al., 2005). PCR reactions were performed in 20 µl, containing 1 ng of DNA, 1x CoralLoad PCR buffer, 3 mM MgCl2, 66 µM of each dNTP, 0.15 µM of each primer, and 0.5 units of Taq DNA polymerase (Qiagen). The amplification protocol was 4 min at 94 °C, followed by 40 cycles at 94 °C for 30 s, 48 °C for 40 s, and 72 °C for 50 s, with a final extension at 72 °C for 7 min. PCR products were purified (Ampure XP Kit; Beckman Coulter) and sequenced in both directions on a 3730xl DNA Analyzer 96-capillary sequencer (Applied Biosystems). We used CodonCode Aligner version 3.7.1 software (CodonCode Corporation, Dedham, MA, USA) to edit sequences, which were 670 bp in length, compared them to the GenBank database content with BLAST, and deposited them in GenBank under accession numbers KT023566, KT023567, KT023568 and KU739501 –KU739517. Species identification was confirmed with the BOLD identification engine (Ratnasingham & Hebert, 2007).

Monogeneans

The host specimens of H. haifensis were not in a perfect state of freshness and the monogeneans were not alive when they were collected. We used seawater to rinse parasites from host gills into Petri dishes, and we further isolated them under a stereomicroscope with incident lighting to prepare them for additional microscopic evaluation. The majority of specimens were mounted in Berlese fluid (hereafter designated ‘b’), a technique which flattens the specimens. A few unflattened monogeneans were dehydrated in an ethanol series, stained with carmine, cleared with clove oil and mounted in Canada balsam (hereafter ‘uc’).

Most monogeneans collected from H. haifensis belonged to a single, abundant, species of Pseudorhabdosynochus (Table 1); the very few specimens from other species are noted but not otherwise considered here.

For illustration of parasites we used an Olympus BH2 microscope equipped with drawing apparatus and differential interference contrast (DIC) optics. The measurements of sclerotised parts, all in micrometres, were taken with the help of a custom-made transparent ruler and are expressed as the range followed in parentheses by the mean, the standard deviation when n ≥ 29, and (n) the number of observations; measurements were taken as in Fig. 1 in Chaabane, Neifar & Justine (2015). The measurements of the right-hand haptoral hard-parts and left-hand equivalents were pooled. Because measured lengths may vary as a function of how specimens are prepared and the degree to which they are flattened (Justine, 2005b), here they are given separately for specimens prepared, respectively, in Berlese (b) and carmine (uc). The terminology for different parts of the male quadriloculate organ and the vagina is that of Justine (2007a). We scanned drawings and used Adobe Illustrator software (version CS5) to refine lines and in some cases to add colour fill to better graphically differentiate structural elements. Museum abbreviations used are as follows: MNHN, Muséum National d’Histoire Naturelle, Paris; BMNH, Natural History Museum, London.

COI sequences of monogeneans

We used a QIAmp DNA Micro Kit (Qiagen) to extract DNA from a whole monogenean specimen (from fish Hh4; Table 1). The specific primers JB3 (=COI-ASmit1) (forward 5′-TTTTTTGGGCATCCTGAGGTTTAT-3′) and JB4.5 (=COI-ASmit2) (reverse 5′-TAAAGAAAGAACATAATGAAAATG-3′) were used to amplify a fragment of 424 bp of the COI gene (Bowles, Blair & McManus, 1995; Littlewood, Rohde & Clough, 1997). PCR reaction was performed in 20 µl, containing 1 ng of DNA, 1x CoralLoad PCR buffer, 3 mM MgCl2, 0.25 mM dNTP, 0.15 µM of each primer, and 0.5 units of Taq DNA polymerase (Qiagen). Thermocycles consisted of an initial denaturation step at 94 °C for 2 min, followed by 37 cycles of denaturation at 94 °C for 30 s, annealing at 48 °C for 40 s, and extension at 72 °C for 50 s. The final extension was conducted at 72 °C for 5 min. Sequences were edited with CodonCode Aligner software version 3.7.1 (CodonCode Corporation, Dedham, MA, USA), compared to the GenBank database content with BLAST, and deposited in GenBank under accession number KT023569.

Trees and distances

A tree was constructed from all available COI sequences of species of the genus Hyporthodus, including sequences already available in GenBank and our new sequences. The tree was inferred using Maximum Likelihood method. The best evolutionary model for the data set was estimated in MEGA7 (Kumar, Stecher & Tamura, in press) under the Bayesian Information Criterion (BIC) to be Hasegawa–Kishino–Yano model (Hasegawa, Kishino & Yano, 1985) with a discrete Gamma distribution (HKY + G). The tree was computed in MEGA7, with 100 bootstrap replications. A tree inferred from the same data, using the Neighbour-Joining method (Saitou & Nei, 1987) and evolutionary distances computed using the Kimura-2 parameter (Kimura, 1980) with 1,000 bootstrap replicates, was also constructed with MEGA7. Genetic distances (Kimura-2 parameter distance) were estimated with MEGA7. All codon positions were used.

Results

Identification of fish hosts, Haifa grouper

When we began our study, no sequence of H. haifensis was available in GenBank, and identification of our first COI sequences via BOLD yielded confusing results, probably because of sequences in the database derived from misidentified specimens. We obtained COI sequences for one specimen from Tunisia (now deposited in the MNHN collections) and two additional specimens from Libya, and the three sequences were identical or very similar (1 bp difference); sequences were also identical or very similar (1 bp difference) to three sequences of H. haifensis (KJ709537, KJ709538 and KJ709539) recently added to GenBank (Landi et al., 2014), from off Sicily, i.e., geographically close to Tunisia and Western Libya. We conclude with certainty, from identical COI sequences and convergent morphological identification, that our fish specimens belong to the species H. haifensis.

Comparison of barcode sequences from Hyporthodus species

We obtained 8 and 9 COI new sequences from H. niveatus and H. nigritus, respectively. In both cases, these sequences were similar to or identical with sequences deposited under the same names in GenBank. A ML analysis (Fig. 1) produced distinct branches for the species H. octofasciatus, H. haifensis, H. acanthistius, H. niveatus, H. nigritus and H. flavolimbatus; H. ergastularius and H. septemfasciatus were not well resolved, but this might be due to misidentification of some sequences, as previously suggested (Schoelinck et al., 2014); a NJ bootstrap analysis produced the same tree topology (Fig. 1). With the exception of the two latter species, all species, and especially H. haifensis, H. nigritus and H. niveatus, were each in separate clades with high support (100%). Specimens of H. nigritus and of H. niveatus were grouped with specimens previously identified under the same names (a single sequence in the case of H. nigritus, 6 sequences in the case of H. niveatus). Hyporthodus haifensis was not closely related neither to H. niveatus (5.6–6% distance) nor to H. nigritus (6.8–7% distance), and the three species were not sister-species (Fig. 1); however, precise phylogenetic relationships between the three species could not be determined because of very low support of several nodes in the phylogenetic analysis.

Figure 1 Tree of Hyporthodus spp. based on COI sequences.

The tree was constructed using the Maximum Likelihood method (100 replicates); a tree constructed using the Neighbour-Joining method (1,000 bootstrap replicates) showed the same topology except for some minor differences in the basal, non-Hyporthodus, branches; the NJ tree is shown. Support for major nodes is indicated for the two methods (as: ML/NJ). The scale bar indicates the number of substitutions per site (ML). The three species involved in our study, namely Hyporthodus haifensis, H. niveatus and H. nigritus, showed independent clades with 100/100 support. However, some higher nodes have low support.

Morphology of monogeneans: Pseudorhabdosynochus sulamericanus Santos, Buchmann & Gibson, 2000

• Taxonomic summary

Synonym: Pseudorhabdosynochus sp. of Chaabane, Neifar & Justine, 2015.

Type-host: Hyporthodus niveatus (Valenciennes, 1828).

Type-locality: Off Brazil (Santos, Buchmann & Gibson, 2000).

Other hosts: Hyporthodus nigritus (Holbrook, 1855) (Kritsky, Bakenhaster & Adams, 2015); Hyporthodus haifensis (Ben-Tuvia, 1953) (this paper).

Other localities: Off Florida (Kritsky, Bakenhaster & Adams, 2015); off Sfax, Tunisia, and Tripoli, Libya (this paper).

Infection site: Gill lamellae.

Prevalence: In our specimens from Tunisia and Libya, 4/4 (100 %), see Table 1.

Material examined: 343 voucher specimens from H. haifensis from off Tunisia and Libya (Table 1), MNHN HEL555; 2 paratypes from H. niveatus off Brazil (BMNH 1999.1.6.1-3); 2 voucher specimens from H. niveatus off Florida (MNHN HEL459, HEL460).

• Redescription (Figs. 2– 6)

Redescription (based on 36 specimens in Berlese and 18 unflattened specimens in carmine from H. haifensis from off Tunisia and Libya; for measurements of other specimens, see Table 3). Adult length uc 634 (500–800, n = 14), b 727 (350–980, n = 16) long, including haptor; maximum width uc 182 (100–270, n = 14), b 230 (115–310, n = 16) at level of ovary (Fig. 2A). Tegument scaly in posterior region (Fig. 6F). Anterior region with 3 pairs of head organs and 2 pairs of dorsal eye-spots, distance between outer margins of anterior eye-spots uc 26 (20–29, n = 7), b 30 (25–38, n = 5), of posterior eye-spots uc 31 (18–39, n = 12), b 34 (28–42, n = 7). Pharynx medial, subspherical. Oesophagus very short or absent. Two simple lateral intestinal caeca not united posteriorly. Haptoral peduncle present. Haptor trapezoidal, width uc 181 (150–200, n = 8), b 208 (180–240, n = 6). Dorsal squamodisc, length uc 84 (75–100, n = 10), b 99 (85–120, n = 13), width uc 75 (59–90, n = 10), b 104 (70–122, n = 13) (Fig. 6D). Ventral squamodisc, length uc 89 (73–150, n = 11), b 100 (78–120, n = 15), width uc 89 (73–150, n = 11), b 100 (78–120, n = 15) (Figs. 5F and 6C). Squamodiscs with 15–16 concentric rows of rodlets; 1 innermost row u-shaped. Rodlets with visible spurs (‘éperons’) (Figs. 6C and 6D). Ventral anchors with handle and distinct guard, outer length uc 44 (40–48, n = 6), b 49 ± 3.1 (40–54, n = 58), inner length uc 41 (30–47, n = 4), b 44 ± 3.1 (32–50, n = 54) (Figs. 5B and 6E). Dorsal anchors with indistinct guard, outer length uc 41 (35–45, n = 7), b 44 ± 2.6 (36–48, n = 51), inner length uc 29 (25–31, n = 3), b 29 ± 2.8 (24–36, n = 33) (Figs. 5C and 6E). Lateral dorsal bars, with flattened medial end, length uc 60 ± 2.3 (55–65, n = 29), b 82 ± 9.3 (60–115, n = 67), maximum width uc 22 ± 3.2 (15–28, n = 29), b 30 ± 4.4 (18–38, n = 67) (Figs. 5E and 6E). Ventral bar long, sometimes V-shaped, with constricted median portion, length uc 93 (82–120, n = 12), b 118 ± 11 (88–135, n = 29), maximum width, uc 17 (13–28, n = 12), b 20 ± 4. 2 (13–26, n = 30) (Figs. 5A, 5D and 6E); for V-shaped ventral bars, measurements were taken as in Fig. 5D.

Figure 2 Pseudorhabdosynochus sulamericanus from Hyporthodus haifensis.

(A) composite, ventral view; tegumental scales not drawn. (B, C) male quadriloculate organ. (D) sclerotised vagina. (A, C) carmine; (B, D) Berlese.

Table 3 Pseudorhabdosynochus sulamericanus.

Comparison of measurements and counts taken from specimens of various origins.

Source	Santos, Buchmann & Gibson (2000) Original description	Kritsky, Bakenhaster & Adams (2015)	Paratypes BMNHN Slides1999.1.6.1-3	Vouchers MNHN Slides HEL460 HEL459	Present study MNHN slides HEL555	
Hosts	H. niveatus	H. nigritus	H. niveatus	H. niveatus	H. niveatus	H. haifensis new host record	
Locality	Off Ilhas Cagarras, Rio de Janeiro, Brazil	Off Florida	Off Florida	Off Ilhas Cagarras, Rio de Janeiro, Brazil	Off Florida	Sfax, Tunisia Tripoli, Libya	
Method	Gomori’s trichrome, Mayer’s paracarmine	Gomori’s trichrome Gray and Wess medium	Gomori’s trichrome Gray and Wess medium	Mayer’s paracarmine	Gray and Wess medium	Gomori’s trichrome	Berlese	Carmine	
Measurements									
Body length	598–1,100 (n = 11)	879–880 (n = 1)	542 (460–649, n = 21)	900 (n = 2)	560	530	727 (350–980, n = 16)	634 (500–800, n = 14)	
Body width	169–228 (n = 11)	179–180 (n = 1)	170 (137–201; n = 22)	190 (180–200, n = 2)	205	150	230 (115–310, n = 16)	182 (100–270, n = 14)	
Haptor width	–	165–166 (n = 1)	160 (131–180, n = 21)	195 (190–200, n = 2)	205	140	208 (180–240, n = 6)	181 (150–200, n = 8)	
Pharynx length	34–52 (n = 11)	–	–	52 (48–55, n = 2)	37	31	–	38 (27–45, n = 15)	
Pharynx width	29–43 (n = 11)	45–46 (n = 1)	38 (34–43, n = 22)	52 (48–56, n = 2)	30	32	–	37 (29–45, n = 15)	
Penis internal length	–	–	–	63 (61–65, n = 2)	–	55	71 ± 6.5 (56–82, n = 32)	49 (45–59, n = 17)	
Penis cone length	–	–	–	6 (5–6, n = 2)	8	5	5 ± 1.1 (4–10, n = 31)	5 (5–7, n = 17)	
Penis tube length	–	–	–	16 (14–17, n = 2)	15	13	14 ± 1.1 (12–17, n = 30)	13 (10–17, n = 16)	
Penis tube diameter	–	–	–	4 (4–4,5, n = 2)	4	3.5	4 ± 0.6 (3–5, n = 31)	4 (3–4, n = 16)	
Penis filament length	–	–	–	2 (0–3, n = 2)	5	4	4 ± 1.7 (0–7, n = 29)	3 (2.5–5, n = 15)	
Penis (chamber + cone) length	48–71 (n = 11)	74–75 (n = 1)	71 (65–79, n = 28)		–	–	–	–	
Sclerotised vagina total length	23–27 (n = 11)	–	–	32 (29–34, n = 2)	31	26	35 ± 2.9 (30–42, n = 35)	28 (23–31, n = 4)	
Squamodisc length	76–96 (n = 11)	47–48 (n = 1)	72 (61–79, n = 18)	91 (85–94, n = 4)	83 (80–85, n = 2)	–	101 (70–120, n = 27)	87 (73–150, n = 20)	
Squamodisc width	62–92 (n = 11)	80–81 (n = 1)	71 (63–81, n = 21)	64 (13–90, n = 4)	87 (86–88, n = 2)	–	105 (75–120, n = 27)	80 (59–90, n = 20)	
Squamodisc, number of rows	15–16 (n = 11)	–	14–17 (usually 15)	16 (15–17, n = 2)	16 (15–16, n = 2)	–	15–16	15–16	
Squamodisc, number of closed rows	1	1	1	1	1	–	1	1	
Ventral anchor outer length	39–43 (n = 11)	48 (47–50, n = 5)	41 (38–45, n = 17)	46 (44–50, n = 4)	48 (n = 2)	41	49 ± 3.1 (40–54, n = 58)	44 (40–48, n = 6)	
Ventral anchor inner length	–	–	–	42 (40–46, n = 4)	41 (40–41, n = 2)	38	44 ± 3.1 (32–50, n = 54)	41 (30–47, n = 4)	
Dorsal anchor outer length	41–48 (n = 11)	47 (46–49, n = 5)	40 (38–43, n = 18)	40 (38–42, n = 4)	41 (40–41, n = 2)	36	44 ± 2.6 (36–48, n = 51)	41 (35–45, n = 7)	
Dorsal anchor inner length	–	–	–	25 (24–28, n = 4)	26 (25–26, n = 2)	24	29 ± 2.8 (24–36, n = 33)	29 (25–31, n = 3)	
Ventral bar length	80–96 (n = 11)	83 (80–87, n = 5)	88 (82–97, n = 14)	98 (94–102, n = 2)	92	82	118 ± 11 (88–135, n = 29)	93(82–120, n = 12)	
Ventral bar width	–	–	–	16 (13–19, n = 2)	18	18	20 ± 4. 2 (13–26, n = 30)	17 (13–28, n = 12)	
Lateral bar length	50–71 (n = 11)	65 (58–69, n = 6)	60 (52–65, n = 18)	64 (63–65, n = 4)	62 (6–63, n = 2)	53 (n = 2)	82 ± 9.3 (60–115, n = 67)	60 ± 2.3 (55–65, n = 29)	
Lateral bar width	–	–	–	18 (13–23, n = 4)	23 (22–23, n = 2)	23 (n = 2)	30 ± 4.4 (18–38, n = 67)	22 ± 3.2 (15–28, n = 29)	

Testis subspherical, posterior, intercaecal. Male copulatory organ quadriloculate, first (anterior) chamber as sclerotised as the three others; fourth chamber forming short cone, prolonged by thin sclerotised tube and filament (Figs. 2B, 2C, 6A and 6B). Inner length uc 49 (45–59, n = 17), b 71 ± 6.5 (56–82, n = 32). Cone length uc 5 (5–7, n = 17), b 5 ± 1.1 (4–10, n = 31). Tube length uc 13 (10–17, n = 16), b 14 ± 1.1 (12–17, n = 30); tube diameter uc 4 (3–4, n = 16), b 4 ± 0.6 (3–5, n = 31). Filament with extremity often bifid, length uc 3 (2.5–5, n = 15), b 4 ± 1.7 (0–7, n = 29).

Figure 3 Pseudorhabdosynochus sulamericanus from various hosts, structure of sclerotised vaginae.

(A, F) specimens from H. niveatus, Brazil, paratypes, BMNH 1999.1.6.1-3. (C) specimen from H. niveatus, Florida, voucher MNHN HEL460. (B, D, E, G) specimens from Hyporthodus haifensis, Libya, vouchers MNHN HEL555. Flattening and staining: (B, E, G) Berlese; (D) carmine; (A, F) trichrome-carmine; (C) Gray and Wess medium.

Vitelline follicles lateral, coextensive with intestinal caeca and contiguous posteriorly to testis. Ovary on right side, looping dorsoventrally around right intestinal caecum. Eggs observed within genital ducts reniform, with thickest shell at proximal pole, polar filament absent, length b 120–128 (n = 2), width b 40–43 (n = 2).

Sclerotized vagina consists of slightly sclerotised funnel-shaped trumpet, followed by short primary canal with thick wall (Figs. 2D, 3 and 4). Primary canal surrounded by additional sclerotised material in its proximal part, which obscures internal relationships. Posterior end of primary canal directed to primary chamber, junction between two structures visible (in specimens from H. niveatus) or not (in specimens from H. haifensis). Primary chamber small, pear-shaped. Secondary canal (junction between primary chamber and secondary chamber) not seen. Secondary chamber spherical, heavily sclerotised. Accessory structure with internal canal, looping twice, inserted on secondary chamber. Total length of sclerotised vagina uc 28 (23–31, n = 4), b 35 ± 2.9 (30–42, n = 35). Diameter of secondary chamber uc 6 (6–7, n = 4), b 5 ± 0.5 (4–6, n = 36). In specimens from Hyporthodus niveatus, the structure is identical but the continuity from the primary canal to the primary chamber could be followed, in contrast with specimens from H. haifensis.

• Remarks on morphology

Most authors have emphasized the importance of the morphological structure of the sclerotised vagina for Pseudorhabdosynochus species identification (Chaabane, Neifar & Justine, 2015; Justine, 2005a; Justine, 2005b; Justine, 2007a; Justine, 2007b; Justine, 2008c; Justine, 2010; Justine, Dupoux & Cribb, 2009; Justine et al., 2010; Justine & Sigura, 2007; Knoff et al., 2015; Mendoza-Franco, Violante-González & Herrera, 2011; Neifar & Euzet, 2007), although the quadriloculate organ and the hard parts of the host attachment apparatus (haptor) including the squamodisc are additional characters for species diagnosis.

Figure 4 Homologies of various parts of the sclerotised vagina of Pseudorhabdosynochus sulamericanus compared to a general diagram.

Colours are similar in homologous parts. The junction between primary canal and primary chamber was not visible in specimens from Hyporthodus haifensis but was seen in specimens from H. niveatus. The secondary canal (junction between primary chamber and secondary chamber) was not visible in any specimen. General diagram adapted from Justine (2007a).

Several Pseudorhabdosynochus species have in common with P. sulamericanus the following vaginal characters: a wide and visible trumpet; diameter of secondary chamber clearly larger than that of primary chamber. These species are: P. dolicocolpos Neifar & Euzet, 2007, P. enitsuji Neifar & Euzet, 2007, P. morrhua Justine, 2008, and P. firmicoleatus Kritsky, Bakenhaster & Adams, 2015.

- P. dolicocolpos (from Mycteroperca costae off Tunisia and Senegal) has a long, coiled thin-walled primary canal (vs short, straight and sclerotised in P. sulamericanus); although the structure is similar, the general shape of the sclerotised vagina is very different. In addition, its male copulatory organ has a long tube (35–45 vs 10–17) (Neifar & Euzet, 2007).

- P. enitsuji (from M. costae off Tunisia and Senegal) has a less conspicuous trumpet and a well-visible primary canal. In addition, its male copulatory organ has a long tube (55–70 vs 10–17) (Neifar & Euzet, 2007).

- P. morrhua (from M. morrhua off New Caledonia) has a less conspicuous trumpet and a thin-walled primary canal (vs sclerotised). In addition, the anterior chamber of its male copulatory organ has a very thin wall (vs as sclerotised as other chambers in P. sulamericanus) (Justine, 2008c).

- P. firmicoleatus (from H. flavolimbatus, type-host, and H. niveatus, both off Florida) was considered as closely resembling P. sulamericanus (Kritsky, Bakenhaster & Adams, 2015). However, Kritsky, Bakenhaster & Adams (2015) enumerated several morphological differences between the two species: absence of additional structure around the sclerotised vagina in P. firmicoleatus (vs present in P. sulamericanus), anchor morphology, tegumental scales (lacking in P. firmicoleatus) and number of rows of rodlets in the squamodisc (12 (11–13) in P. firmicoleatus vs 15 (14–17) in P. sulamericanus).

In none of these three species is there additional sclerotised material around the primary canal of the sclerotised vagina, as in P. sulamericanus. In P. sulamericanus, the male quadriloculate organ has the usual structure found in species of Pseudorhabdosynochus, but a minor difference can be detected at its distal extremity, i.e., a thin and short filament with bifid extremity. However, this detail itself could not be considered alone as a differential character for the species because it is variable; it was not mentioned in the original description or redescription (Kritsky, Bakenhaster & Adams, 2015; Santos, Buchmann & Gibson, 2000).

Pseudorhabdosynochus sulamericanus has an exceptional vaginal structure. In most Pseudorhabdosynochus species, there is a general pattern in which the continuity of the lumen can be followed from trumpet to secondary chamber through primary canal, primary chamber and secondary chamber (Justine, 2007a). This continuity is likely to correspond to the complex journey of inseminated sperm through the female organ, from the entrance (trumpet) to the secondary chamber which exits into a soft tube connected to the oötype (Justine, 2009). In specimens of P. sulamericanus from H. haifensis, we could discern neither the continuity between the primary canal and the primary chamber, nor the continuity from the primary chamber to the secondary chamber through the secondary canal. This is probably due to the presence of the additional sclerotised material which obscures vision. However, in specimens from H. niveatus, the primary canal—primary chamber continuity could be seen, but we evaluated far fewer of these specimens and so cannot be sure about the range of morphological variability in this structure. The additional sclerotised material is visible in the drawings of the original description and probably mentioned as “enclosed in muscular, funnel-shaped organ” (Santos, Buchmann & Gibson, 2000); it is mentioned in its redescription as “surrounded by variable small sclerites” (Kritsky, Bakenhaster & Adams, 2015); none of these authors used a DIC microscope which provides a better resolution of the hollow sclerotised organs.

Figure 5 Pseudorhabdosynochus sulamericanus from Hyporthodus haifensis, haptor hard parts and squamodisc.

(A, D) ventral bar, with method of measurement of length; (B) ventral anchor; (C) dorsal anchor; (E) lateral (dorsal) bar; (F) ventral squamodisc. All Berlese.

The only other species of Pseudorhabdosynochus found on species of Hyporthodus are P. querni (Yamaguti, 1968) Kritsky & Beverley-Burton, 1986 from H. quernus off Hawaii, and P. firmicoleatus from H. flavolimbatus and H. niveatus, both off Florida. Pseudorhabdosynochus querni has a vaginal structure very different from that of P. sulamericanus (Yamaguti, 1968; Yang, Gibson & Zeng, 2005); P. firmicoleatus has a somewhat similar vaginal structure (Kritsky, Bakenhaster & Adams, 2015) but can also be distinguished by other characteristics (see above).

COI sequences of monogeneans

We obtained COI sequences of P. sulamericanus from H. haifensis from off Tunisia and Libya. The closest sequence in GenBank according to BLAST was from P. cyanopodus Sigura & Justine, 2008 (Schoelinck, Cruaud & Justine, 2012), a parasite from Epinephelus spp. in the South Pacific. The sequences differed by 17.6% (Kimura-2 parameter distance). Since no sequence of P. sulamericanus from the Americas was available, no further comparison was possible.

Figure 6 Pseudorhabdosynochus sulamericanus from Hyporthodus niveatus, male copulatory organ, haptor hard parts, squamodiscs.

(A, B) male copulatory organ; (C, D) squamodiscs (C, ventral; (D, dorsal); (E) haptoral parts, (F) tegumental scales. (A) MNHN HEL459, from Florida, Gomori, unflattened; (B) BMNH 1999.1.6.1-3, from Brazil, trichrome carmine; (C, D, E, F) MNHN HEL460, from Florida, Gray and Wess medium.

Discussion

Based on our observations on specimens collected in the Mediterranean and museum specimens, the same species, P. sulamericanus, is found on different species of groupers, one, Hyporthodus haifensis, in the eastern Atlantic (including the Mediterranean Sea) and two, H. niveatus and H. nigritus, in the western Atlantic (including the Gulf of Mexico) (Fig. 7). These congeneric fishes are all considered deep-water species, and as adults none of them typically ranges into water shallower than 55 m (Froese & Pauly, 2016), a trait making them logistically difficult to observe and collect. Hyporthodus Gill is a genus that was recently resurrected on the basis of molecular data (Craig & Hastings, 2007), for a monophyletic group of deep-groupers previously classified within Epinephelus Bloch; morphological differentiation of this genus is possible from a unique arrangement of the coracoid and cleithrum and position of pelvic fins (Craig, Sadovy de Mitcheson & Heemstra, 2012) and the monophyly of the genus was confirmed in a recent molecular study (Schoelinck et al., 2014). Our phylogenetic analysis, based on COI sequences showed that the three species H. haifensis, H. niveatus and H. nigritus are distinct, with distances between species ranging 5.6–7%, whereas intraspecific COI distances in groupers are reported as 0.7–4% (Alcantara & Yambot, 2014). In our analysis, H. haifensis is not closely related to H. niveatus and H. nigritus, and, in the context of available COI sequences and low support for several nodes in our analysis, none of the three species is sister-species of one of the others (Fig. 1), so phylogenetic similarity does not explain why they would share a putatively host-specific parasite.

Figure 7 Geographical distribution of three species of Hyporthodus in the Atlantic Ocean and Mediterranean Sea, and localities where specimens of Pseudorhabdosynochus sulamericanus were collected.

Hyporthodus haifensis is only known from the Mediterranean Sea and African coasts of the Eastern Atlantic; H. niveatus and H. nigritus are American species. The distributions of the American and African species do not overlap, and are separated by the span of the Atlantic Ocean (Heemstra & Randall, 1993).

It is intriguing that the same species of Pseudorhabdosynochus was found in different species of fish from two sides of the Atlantic. More than 80 species of Pseudorhabdosynochus are known; they are generally extremely species-specific, i.e., a species is found only on one species of host (Justine, 2005a; Justine, 2005b; Justine, 2007a; Justine, 2007b; Justine, 2008b; Justine, 2008c; Justine et al., 2010; Justine & Sigura, 2007; Sigura & Justine, 2008); however, Schoelinck, Cruaud & Justine (2012) recently demonstrated, on morphological and molecular bases, that P. cyanopodus occurs on two sympatric species of deep-sea groupers that inhabit the outer slope off the barrier reef of New Caledonia, South Pacific. These are Epinephelus cyanopodus and E. chlorostigma (Schoelinck, Cruaud & Justine, 2012). Those authors hypothesized that low specificity was an adaptation of P. cyanopodus to deep-sea conditions, where hosts are rare and separated by wide areas, and that infesting two species of hosts helps in perpetuating the parasite species (Schoelinck, Cruaud & Justine, 2012). This hypothesis was coherent with the observation that the species richness of gill monogeneans is five times higher in surface fish than in deep-sea fish (Rohde, 1988). For P. sulamericanus, which parasitizes three species of Hyporthodus that are deep-sea, demersal groupers, the same hypothesis could be proposed for the origin of its low specificity. However, a striking difference between P. sulamericanus and P. cyanopodus is that the hosts of the former are not sympatric, but widely separated by the Atlantic Ocean.

Two other cases of trans-Atlantic species of Pseudorhabdosynochus are found in the literature: they are P. americanus (Price, 1937) Kritsky & Beverley-Burton, 1986 and P. beverleyburtonae (Oliver, 1984) Kritsky & Beverley-Burton, 1986.

In the case of P. americanus, Kritsky, Bakenhaster & Adams (2015) unambiguously demonstrated that previous records on the Eastern side were erroneous and/or based on inadequate synonymies, and concluded that P. americanus was found only on its type-host, the atlantic goliath grouper E. itajara, on the Western side of the Atlantic. Although this grouper is a trans-Atlantic species, no record of P. americanus is known from fish caught on the Eastern side.

Pseudorhabdosynochus beverleyburtonae (Oliver, 1984; Oliver, 1987; Oliver, 1992; Santos, Buchmann & Gibson, 2000) was first recorded from the Mediterranean Sea (under various synonymous names, see Kritsky, Bakenhaster & Adams, 2015) on its type-host the dusky grouper Mycteroperca marginata (synonym Epinephelus marginatus), found in several localities in the Mediterranean (references in Kritsky, Bakenhaster & Adams, 2015) on the same host, then found off Brazil (Kritsky, Bakenhaster & Adams, 2015; Roumbedakis et al., 2013; Santos, Buchmann & Gibson, 2000), each time on the same host species. Kritsky, Bakenhaster & Adams (2015) compared specimens from both sides of the Atlantic, did not find any morphological features that distinguished specimens from these localities, and concluded, as did Santos, Buchmann & Gibson (2000), that the specimens were conspecific. In contrast to our findings for P. sulamericanus, in that case the hosts were also conspecific (M. marginata).

Including E. itajara and M. marginata, there are four species of grouper with trans-Atlantic distribution; the others being the rock hind, E. adscensionis, and the Atlantic creolefish, Paranthias furcifer. Epinephelus adscensionis harbours P. monaensis Dyer, Williams & Bunkley-Williams, 1994 and P. williamsi Kritsky, Bakenhaster & Adams, 2015, both described from specimens collected off Puerto Rico (Dyer, Williams & Bunkley-Williams, 1994; Kritsky, Bakenhaster & Adams, 2015); no record of these species is known from the Atlantic coast of Afro-Eurasia. Kritsky, Bakenhaster & Adams (2015) pointed out that the type-host of P. bocquetae (Oliver & Paperna, 1984) Kritsky & Beverley-Burton, 1986, a species described from the Red Sea and allegedly from this fish, could not be E. adscensionis. Therefore, there is no valid record of Pseudorhabdosynochus species from E. adscensionis on the Eastern side of the Atlantic. The Atlantic creolefish, Paranthias furcifer is not known as a host of any Pseudorhabdosynochus species (Kritsky, Bakenhaster & Adams, 2015).

We searched the literature for records of the same species of monogeneans on both sides of the Atlantic in tropical and warm temperate waters (Table 4). Basically, we used the recent and comprehensive list of monogeneans from South American (Cohen, Justo & Kohn, 2013) and searched the literature for mentions of the same species on the Eurafrican coast. We did not consider fish from subpolar or polar waters because they represent distinct northern and southern populations that are each, respectively, circumglobally homogenous (Froese & Pauly, 2016). Curiously, we found no more than a dozen species, although more than 600 fish monogenean species were listed from South America alone (Cohen, Justo & Kohn, 2013). We noted that no molecular work was undertaken for any of these cases of trans-Atlantic monogeneans. As could be expected, most cases (eight species) concern parasites of pelagic fish with wide distribution, such as Scombridae (tunas) and Clupeidae (sardines); some of these monogeneans were found, not only on both sides of the Atlantic, but also in the Pacific (Table 4). Of these cases, seven are polyopisthocotylean monogeneans, a group of large species associated with these fish families, and which often show wide host specificity; but in at least two of these polyopisthocotylean species, the conspecificity of the American and European forms have been questioned (notes under Table 4). One case is a capsalid (monopisthocotylean) from tunas. Three cases concern sparid fish (Sparidae); two are polyopisthocotylean species for which specimens from both sides of the Atlantic have been comparatively studied (Santos, Souto-Padrón & Lanfredi, 1996). The third case is a diplectanid, Lamellodiscus baeri Oliver, 1974, from Pagrus pagrus in the Mediterranean; since no morphological data are available for its mention in South America (Soares, Vieira & Luque, 2014), we consider that this needs verification. Finally, two cases are diplectanids from groupers: P. beverleyburtonae, which, based on comparative morphological studies, seems be present on both sides of the Atlantic on the same fish, the Dusky grouper (see above); the other is P. sulamericanus, the subject of our study. P sulamericanus is thus unique in that it is the single monopisthocotylean monogenean found on both sides of the Atlantic (The Americas and central Afro-Eurasia) on different species of fish.

Table 4 Species of monogeneans recorded on both sides of the Atlantic.

Group, Family	Species	Western side, South America: Locality, Hosts, references	Eastern side: Locality, Hosts, references	Comments	
Parasites of Scombridae (Tunas, Mackerels): pelagic fish, often with wide distribution or circumglobal	
Monop.; Capsalidae	Nasicola klawei (Stunkard, 1962)	Brazil; Thunnus albacares (Cohen, Justo & Kohn, 2013)	European waters; Thunnus albacares (Gibson, 2016)	Same fish on both sides—Pelagic fish	
Polyop.; Gotocotylidae	Gotocotyla acanthura (Parona & Perugia, 1896)	Brazil; Cynoscion leiarchus, Pomatomus saltatrix (Cohen, Justo & Kohn, 2013)	Many localities, many hosts (Hayward & Rohde, 1999a)	Different fish on both sides of the Atlantic, also in Pacific —Pelagic circumglobal fishes	
Polyop.; Hexostomatidae	Hexostoma auxisi Palombi, 1943	Brazil; Auxis thazard (Cohen, Justo & Kohn, 2013)	Mediterranean Sea; Auxis thazard (Yamaguti, 1963)	Same fish on both sides—Pelagic fish	
Polyop.; Mazocraeidae	Grubea cochlar Diesing, 1858	Brazil, Venezuela; Scomber colias (Cohen, Justo & Kohn, 2013)	Europe, Mediterranean; Scomber scombrus, S. colias (Yamaguti, 1963)	Various fish of genus Scomber on both sides— Pelagic fisha	
Polyop.; Mazocraeidae	Kuhnia scombri (Kuhn, 1829)	Argentina, Brazil, Venezuela; Scomber colias (Cohen, Justo & Kohn, 2013)	Atlantic, Mediterranean, Pacific; various Scomber spp (Yamaguti, 1963)	Various fish of genus Scomber on both sides—Pelagic fish	
Polyop.; Mazocraeidae	Pseudanthocotyloides heterocotyle (van Beneden, 1871) Euzet & Prost, 1969	Brazil, Uruguay; Cetengraulis edentulus, Decapterus punctatus, Anchoa marinii, Engraulis anchoita (Cohen, Justo & Kohn, 2013)	Mediterranean, North Atlantic; Sprattus sprattus, Clupea harengus (Rahimian et al., 1999)	Various fish on both sides—Pelagic fish b	
Polyop.; Thoracocotylidae	Scomberocotyle scomberomori (Koratha, 1955)	Brazil; Scomberomorus cavalla (Cohen, Justo & Kohn, 2013)	Western Africa; Various fish of genus Scomberomorus (Hayward & Rohde, 1999b)	Various fish of genus Scomberomorus, records from both sides of the Atlantic and eastern Pacific—circumglobal pelagic fish species	
Polyop.; Thoracocotylidae	Mexicotyle mexicana (Meserve, 1938)	United States to Brazil, many localities; Scomberomorus spp. (Rohde & Hayward, 1999)	Ghana; Scomberomorus tritor (Rohde & Hayward, 1999)	Various fish of the genus Scomberomorus, many records on Western Side, 1 record on Eastern side, also in Eastern Pacific; circumglobal pelagic fish species	
Parasites of Sparidae (sea breams and porgies): Coastal fish	
Polyop.; Microcotylidae	Atriaster heterodus Lebedev & Parukhin, 1968	Brazil; Diplodus argenteus (Santos, Souto-Padrón & Lanfredi, 1996)	Namibia, Mediterranean Sea, Canary Islands; several Diplodus species (Santos, Souto-Padrón & Lanfredi, 1996)	Fishes of genus Diplodus on both sides—coastal fishc	
Polyop.; Microcotylidae	Polylabris tubicirrus (Paperna & Kohn, 1964)	Brazil; Diplodus argenteus (Santos, Souto-Padrón & Lanfredi, 1996)	Mediterranean Sea; various Diplodus species, Sparus aurata (Santos, Souto-Padrón & Lanfredi, 1996)	Fishes of genus Diplodus on both sides—coastal fishc	
Monop.; Diplectanidae	Lamellodiscus baeri Oliver, 1974	Brazil; Pagrus pagrus (Soares, Vieira & Luque, 2014)	Mediterranean Sea, Pagrus pagrus (Oliver, 1974; Amine & Euzet, 2005)	Same fish on both sides—coastal fish—American record needs verification; see text for comments	
Parasites of Epinephelidae (groupers): Coastal or Deep-Sea fish	
Monop.; Diplectanidae	Pseudorhabdosynochus beverleyburtonae (Oliver, 1984) Kritsky & Beverley-Burton, 1986	Brazil; Mycteroperca marginata (Roumbedakis et al., 2013; Santos, Buchmann & Gibson, 2000; Kritsky, Bakenhaster & Adams, 2015)	Mediterranean Sea; Mycteroperca marginata (Euzet & Oliver, 1965; Oliver, 1968; Oliver, 1984; Oliver, 1987)	Same fish on both sides—coastal fish—see text for comments	
Monop.; Diplectanidae	Pseudorhabdosynochus sulamericanus	Brazil, Florida; Hyporthodus niveatus, H. nigritus (Kritsky, Bakenhaster & Adams, 2015; Santos, Buchmann & Gibson, 2000)	Mediterranean Sea; Hyporthodus haifensis; present paper	Different fish on both sides—deep-sea fish—see text for comments	
Notes.

TITLE Monop. Monopisthocotylea

Polyop. Polyopisthocotylea

Names of fish were updated according to FishBase (Froese & Pauly, 2016).

a Yamaguti (1963) noted: owing to the incomplete description by Linton it is not possible to determine the conspecificity of the American and European forms.

b Rahimian et al. (1999) commented that the specimens from off South America were different, therefore suggesting that species identification needed verification.

c Santos, Souto-Padrón & Lanfredi (1996) compared specimens from both sides and the Atlantic.

It thus appears, rather logically, that the South Atlantic Ocean acts as a barrier to monogenean parasites of demersal fish; this barrier should not concern pelagic fish, which might cross the Ocean, but even these cases are not numerous.

The question remains how the same species of parasite, P. sulamericanus, with very low dispersion abilities as most monogeneans, can be found on different species of fish separated by a wide ocean. We considered several hypotheses. (Hypothesis a) Pseudorhabdosynochus sulamericanus was a parasite of the common ancestor of the three grouper species, and the descending parasite species underwent little or no morphological differentiation since the host species were separated; this hypothesis is hampered by the fact that the three groupers, H. haifensis, H. nigritus and H. niveatus, are not sister-species in our phylogenetic analysis. It might be argued, however, that this analysis was based only on COI sequences and that low support was found for several nodes. (Hypothesis b) The three species of Hyporthodus from the American (H. nigritus and H. niveatus) and African (H. haifensis) sides of the Atlantic, currently have unexpected opportunities to exchange parasites, in an unknown zone of sympatry, or had such opportunities in a recent past. Studies of coral reef groupers have shown that infection of adult fish and exchange of monogeneans between different host species occur during spawning aggregations (Sigura & Justine, 2008). We do not suggest that such spawning aggregations, uniting species from both sides of the Ocean, exist for the Atlantic species of Hyporthodus, but we remark that our knowledge of the behaviour and precise distribution of rare deep-sea groupers is certainly far from exhaustive, thus making the second hypothesis at least plausible.

It did not escape our attention that a molecular study of parasites would provide additional data valuable to this study; unfortunately, fresh specimens of P. sulamericanus from the western side of the Atlantic were not available, in spite of our efforts to obtain them from colleagues, thus molecular comparisons of American and Afro-Eurasian material were not possible. A possibility thus remains (Hypothesis c) that P. sulamericanus is in fact a cryptic species, with one species in the Mediterranean, on H. haifensis, and one (or more) species in the western Atlantic on H. niveatus and H. nigritus. We could not eliminate this hypothesis; however, we are reasonably confident that morphological similarities of material from both sides of the Atlantic, particularly the shared characteristic structure of the sclerotised vagina, provide strong enough evidence to support our conclusion that all specimens reported here belong to P. sulamericanus.

Bernard Séret (IRD-MNHN) kindly examined a fish specimen and prepared it for deposition in the MNHN collections, and provided valuable advice. Eileen Harris (NHM, London) kindly helped with NHM specimens. Wiem Boussellaa (FSS, Tunisia) provided a fish specimen. Delane Kritsky (Idaho State University, USA) kindly authorised us to use data from a paper when it was in press. Tissue samples from Gulf of Mexico groupers were collected by fisheries biologists of the Florida Fish and Wildlife Conservation Commission (FWC), and of them we are particularly grateful to Lew Bullock and Chris Bradshaw. Samantha Gray (FWC) initially processed and catalogued Florida tissue samples and data. The views and conclusions contained in this document are those of the authors and should not be interpreted as representing the opinions or policies of the US government or any of its agencies.

Additional Information and Declarations

Competing Interests

Author Contributions

Animal Ethics

DNA Deposition

Data Availability

Jean-Lou Justine is an Academic Editor for PeerJ.

Amira Chaabane and Jean-Lou Justine conceived and designed the experiments, performed the experiments, analyzed the data, contributed reagents/materials/analysis tools, wrote the paper, prepared figures and/or tables, reviewed drafts of the paper.

Delphine Gey performed the experiments, analyzed the data, reviewed drafts of the paper.

Micah D. Bakenhaster contributed reagents/materials/analysis tools, reviewed drafts of the paper.

Lassad Neifar conceived and designed the experiments, contributed reagents/materials/analysis tools, reviewed drafts of the paper.

The following information was supplied relating to ethical approvals (i.e., approving body and any reference numbers):

All animal research was performed on dead fish purchased from commercial merchants, including fishmarkets.

The following information was supplied regarding the deposition of DNA sequences:

GenBank; all registration numbers are provided within the text.

The following information was supplied regarding data availability:

GenBank; all registration numbers are provided within the text.

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
