# Peer review of "Pseudorhabdosynochus sulamericanus (Monogenea, Diplectanidae), a parasite of deep-sea groupers (Serranidae) occurs transatlantically on three congeneric hosts (Hyporthodus spp.), one from the Mediterranean Sea and two from the western Atlantic"

_PeerJ, doi:10.7717/peerj.2233_

## Round 0.1 · original submission · Major Revisions

· Academic Editor

Major Revisions

The three reviewers are very positive as to the value of the findings presented in this ms, so I think it deserves being published, however all of them find some points at which the ms needs to be ameliorated. In consequence, I propose the need of major revisions since I find that some of the recommendations require important changes (which I think will not be very difficult to do but take some time). A part of addressing the punctual comments of the reviewers (giving answers, making changes or justifying why not changes have been done), I think there are two very important points in the reviews that need to be carefully taken into account.

In the first place, at least two of the reviewers find missing information or think that the text needs reorganization. They point to the introduction to include too much of results and especially discussion (in which I totally agree), and on the other hand reviewer 1 proposes some extra issues to be touched in the introduction or the discussion section. Please read carefully reviewers 1 and 3 comments, and modify the introduction, results and discussion sections accordingly.

In the second place, the reviewers also find that the methodology used to infer the phylogeny of fishes is not the most adequate, especially if the authors want it to draw sound conclusions on their relationships and as a consequence on the evolution of the parasites, as they do. In this respect I agree with reviewers in that to infer a phylogeny nowadays probabilistic methods, as Maximum likelihood, are much more reliable than Neighbour-joining, and no doubt that a phylogeny without bootstrap values is not acceptable at all. The authors must follow the reviewers’ advice of, for example, using ML method as implemented in MEGA7, or even try other software (as RAxML-Gui, or RAxML on line, http://embnet.vital-it.ch/raxml-bb/) and include a bootstrap support for the nodes.

Reviewer 1 ·

Basic reporting

L. 58: in these cases, the FishBase consortium, as far as I know, recommends to cite the original reference/source dealing with the fish taxon in question, rather than the FishBase website itself, except in case of e.g. meta-analyses or for general statements e.g. about nomenclature (like on L. 96-97), in which case FishBase should be cited.

Although it does clarify things from the beginning, I find it a little bit strange that relatively much of the findings and discussion topics are already touched upon in the Introduction session. I leave it up to the editor to decide whether overlap between Introduction and Discussion should be less.

L. 382: “hampered BY the fact”, isn’t it?

Experimental design

I’d like some explanation why exactly COI was chosen as a marker to check for sister-group relationships in the host fishes. Of course, barcoding is a widely used approach; on the other hand, a lot of pros and cons can be mentioned regarding the use of one (mitochondrial) marker only to infer phylogenetic relationships, as is also mentioned by the authors on L. 383-384. Could the authors comment on this? This could perhaps also be linked to why COI was chosen as a marker for the monogeneans, too.

L. 90: would it be possible to specify which source/reference was used to identify fish?

L. 106: how were PCR products purified?

L. 147: why was the K2P model of molecular evolution chosen? (I would, by the way, suggest to write “kimura” with an upper-case letter in the beginning when spelled out in full.)

Validity of the findings

L. 174-176: it would be useful to have some idea of interspecific/intraspecific distances over this gene in other serranids, to allow the reader to estimate what kind of distance is typical between closely related species, sister species… as these are otherwise very relative concepts.

Fig. 1: why no bootstrap values on this tree (hence also, why no bootstrapping when building the tree)? I appreciate this is no phylogeny reconstruction (although the presence of an outgroup etc. does suggest the tree intends to show more than just visualising COI distances), so bootstrapping would indeed be somewhat less crucial here, but on the other hand, it would allow the reader to judge to what extent Hyporthodus haifensis sequences form a firm cluster etc.
Also, I’d appreciate some explanation on why and how the outgroup was chosen.

L. 305-306: how well were closely related hosts scrutinised for monogeneans?

Additional comments

The manuscript is well-written and deals with an interesting topic. I enjoyed reading it. I appreciate the careful approach of the material studied, which includes rendering it available for future workers, including e.g. vouchering. Also, the detailed morphological redescription, covering plenty of traits often not covered by many monogenean workers, will prove very useful to future research. Hence, this is a valuable study and I recommend publishing it. I do have some relatively minor suggestions that I’d suggest the authors to address; these are detailed elsewhere in this report.

A general comment that I’d like to make is that it would be interesting, in the Introduction or the Discussion, to devote some additional references/sentences to situating this study in a somewhat wider framework:
1) how does this story compare to other grouper parasites (e.g. do non-monogeneans infecting groupers generally exhibit a wide host range, a wide geographical distribution…)?
2) how does it compare to other (monogenean) parasites of deepwater fishes, other than groupers?
I ask this because I think that the authors are uncovering a really interesting phenomenon and distribution pattern here, and it would in my view render it more palatable to non-specialists if these results are put in a somewhat wider framework in this way.

I hope the authors find my comments useful.

·

Basic reporting

The article is written using appropriate language. The introduction is correct and the literature is well cited. The structure of the article is also correct. Regarding the figures, although all of them are suitable, in my opinion they are not well organized, because I believe that the first figure should be corresponding to the distribution map (which is now as Figure 7). There is an error in the Figure 7 legend: italic letters and normal are interchanged, so that remain species names without being italicized and should be the opposite. Figure 1 also has an error, and that is that the names of the species should be in italics again, both terminals of the branches and the names of the species that are situated in an explanatory way to the right of the figure.

Experimental design

The article defines a clear biological question that is adequately resolved; however, the methodological part is not the most appropriate in some respects. I do not have enough knowledge in the area of morphological description of parasites, so I am not able to discuss much about this part. Anyway, I think it would be much more explanatory and easy to read if the discussion and comparison with other species was done at the same time as the description of the morphological characters is done.
In the part of material and methods should be given more information about the DNA extraction from fish, for example, which tissue has been used for this, or has been done with blood? What is the length of the COI fragment that was amplified? Is it the same as for parasites?
Referring to the molecular part, analyses are somewhat poor at phylogenetic level. Phylogenies lack statistical support (it is not implemented any bootstrap or other test of reliability in order to validate the nodes presented in the trees) and they are only based on genetic distances. It would be appropriate to use more complex methods (such as Maximum Likelihood or Bayesian Inference) based on probabilistic approaches to have more confidence in the results presented. It wasn’t studied what is the evolutionary model that best fits the data, and it is known that the model used in the inference results may vary the phylogeny. Thus, all conclusions drawn from the phylogenetic inferences can not be 100% trusted, as the data were not adequately addressed.

Validity of the findings

When interpreting the results and in order to understand the biological significance of these, it would have been helpful to have more information on hosts, the fish. To try to understand whether these animals of different species have been able to contact at some point in history, dispersion data of these species would have been helpful.
It has been sampled in the intermediate region? It is possible that are missing the links in this so disjunct distribution.
On the other hand, although the authors already mentioned it at the end of the discussion, the work is incomplete, because the ideal situation would have been to have the proper phylogeny of fish and to compare it with the phylogeny of parasites. Firstly to be confident about that it is the same species in all three cases (P. sulamericanus) rather cryptic species (as the authors say in one of their hypotheses) and secondly to study whether parasites have followed a parallel evolution to they have had their hosts. There is no possibility of obtaining material for DNA extraction of parasites? What they exactly mean when they say that no fresh material is available? Haven’t they tried to do DNA extractions from the material they have used to make the morphological description? There exist very sophisticated extraction methods where you can get DNA from nearly any material that contains it. It’s worth a try.
Even if it were not possible to obtain sequences of the parasites from the other hosts, also it could have been inferred a phylogeny with other sequences in GenBank to place the putative P. sulamericanus sequence which has been obtained. Although there is no information in the database of this particular species, there is information about the phylogenetic relationships with other groups (at least at the morphological level) so that it would be possible to place it in a phylogeny and discard the possibility that it belongs to another group.

Additional comments

No Comments.

Reviewer 3 ·

Basic reporting

• The submission adheres to all PeerJ policies.
• The English language (grammar and style) should be revised.
• The article lacks background information, the authors should expand the introduction.
• The structure of the submitted article needs to be revised.
• The figures need to be edited for visual clarity.
• The submission is ‘self-contained’.
• Yes, all appropriate raw data has been made available in accordance with our Data Sharing policy.

Experimental design

• The submission describes original primary research within the Scope of the journal.
• The submission should improve on clarity when defining the research question. The knowledge gap being investigated needs to be clarified in the introduction, and clearer statements need to be made as to how the study contributes to filling that gap.
• The investigation has been conducted adequately but the data analyses need to be improved.
• Some of the methods are missing. The rest of the Methods described are adequate.
• The research seems to have been conducted in conformity with the prevailing ethical standards in the field.

Validity of the findings

• The data is basic, but acceptable for the line of research.
• The data on which the conclusions are based has been made available.
• The conclusions are too far-fetched given the data and should be toned down.
• Speculation has mostly been identified as such, but some of the conclusions remain speculative and this should be made clear.

Additional comments

In continuation, I mention my general opinion and recommendations for this study.
This manuscript provides interesting information about the intraspecific morphological variability of a parasite species with extended geographical distribution, Pseudorhabdosynochus sulamericanus from the Mediterranean Sea and western Atlantic. The morphological variation of distinct morphological traits associated to three hosts of the genus Hyporthodus is discussed as an original idea in the context of describing parasite biodiversity. However, I detected three major problems in the actual version of this MS:

1. The structure of this MS not was prepared following the Instructions for the Authors of PeerJ. Please, check the author guidelines to verify.
2. The authors do not show robust data for inferring the evolutionary hypothesis to explain their findings, because of an apparent limited knowledge on the background of molecular systematics and evolution.
3. The raw data of this study are new DNA sequences of three species of marine fishes of wildlife from Hyporthodus genus. However, the authors do not appear to have carried out the most appropriate analyses for the dataset, apart from only having one gene fragment.
For this reason, my major recommendation for this manuscript is to reorganize the contents and the results, and to tone down the conclusions based on weak data, discussing drawbacks and possible improvements. In addition, the manuscript has several grammatical errors, please have the English checked.
In general, this manuscript has several formatting problems:
1. It is not necessary to mention both the common and scientific names of the marine fish species throughout the manuscript, it is sufficient to mention both names once at the beginning and then continue with either the common, or scientific name for the rest of the text.
2. Please specify the locality and country of each record mentioned in the MS and please remain consistent with the formatting of the style.
3. The authors failed in the order in which each figure appears on the file upload screen. Is necessary to change all the numbers and reorganize the figures based on their appearance in the text.

I understand that PeerJ has a philosophy of “…authors spending their time doing science, not formatting”; however, is important that the authors show an effort to transmit their research. In this context, I suggested the authors revise with special emphasis their writing style. For example, on the several occasions, I found duplicate citations in the same sentence; e.g., Line 274-278: “The additional sclerotised material is visible in the drawings of the original description by Santos et al. 2000 and probably mentioned as “enclosed in muscular, funnel-shaped organ” (Santos et al. 2000); it is mentioned in its redescription by Kritsky et al. as “surrounded by variable small sclerites” (Kritsky et al. 2015); none of these authors used a DIC microscope which provides a better resolution of the hollow sclerotised organs”. It would be better to write: “The additional sclerotised material is visible in the drawings of the original description by Santos et al. (2000) and probably mentioned as “enclosed in muscular, funnel-shaped organ”; it is mentioned in its redescription by Kritsky et al. (2015) as “surrounded by variable small sclerites”; none of these authors used a DIC microscope which provides a better resolution of the hollow sclerotised organs”. The same error is commonly found in several parts of this MS. Please, amend.
In any case, a few recommendations for the authors to consider will be given below.
Introduction.
The aim of this research article is not explicit and should be justified by restructuring the introduction. Also, the introduction as a whole should contain more background information. Furthermore, the introduction contains information that should be separated into Results and Discussion. For example:
Line 66-70. This entire paragraph should be moved to the Results or Discussion sections.
Line 71-73. This sentence is not supported by the results shown in this MS, but can be used as a conclusion and directive for futures studies.
Line 76-77. The authors mention that: “Results of these analyses confirm that these grouper species are distinct and not closely relate”. First, again, this paragraph should be used in results not in the Introduction. Second, to state that these species are “not closely related” is a bold statement to make based on the superficial analyses carried out, as I explain further on.
Line 77-86. This entire section should be in the Discussion.
Materials and Methods.
Line 88. Please change “Fish” to “Collected Fish”.
Line 92. Please provided voucher numbers.
Line 132. Please mention the version of Adobe Illustrator.
Line 133. Please change “MNHN, Museum National d’Histoire Naturelle, Paris; BMNH, Natural History Museum, London.” to “Museum National d’Histoire Naturelle (MNHN), Paris; Natural History Museum (BMNH), London.

Line 137-139. Please change “COI-ASmit1” to “JB3” and “COIASmit2” to “JB4.5.On the other hand, “(Littlewood et al., 1997)” is not cited correct for both primers. Please, cite the correct paper for both primers as: “Bowles J., Blair, D. and D.P. McManus. 1995. A molecular phylogeny of the Schistosomes. Molecular Phylogenetics and Evolution. 4:103-109.”
Line 143-144. Please delete “Aligner software (CodonCode Corporation, Dedham, MA, USA)”, as it was previously mentioned in the text (see Line 107, and in this section mention the version of this Software).
Line 146. Please change “Trees and distances” to “Trees and distances from Hyporthodus genus”.
Although it is common practice to show distance-based trees when making “barcode” comparisons, for the purpose of publications and drawing phylogenetic and systematic conclusions this type of analysis is inappropriate and superficial. In general my main recommendation for this section (Lines 147-149) is to modify the phylogenetic analyses at least to a Maximum Likelihood based approach (for example available in MEGA 7.0. (MEGA 6 has been updated now). This will at least provide the study with a phylogenetic tree containing nodal support values, otherwise the conclusions drawn (i.e. that the three Hyporthodus species from this study are not monophyletic) are meaningless. Also, the genetic distances should be presented as mean interspecific “p-distances” (also implement in MEGA 7.0).
Results.
Lines 151 -165. All this paragraph can be re-located to the discussion section.
Line 174-176. The author mentioned that: “More importantly for our study, H. haifensis was not closely related neither to H. niveatus (5.6-6% distance) nor to H. nigritus (6.8-7% distance), and the three species were not sister-species (Figure 1).
I don’t understand: First. Why “more importantly”? The aim if this paper is not to test whether these species are sister species. Secondly, as I mentioned before, these kinds of conclusions drawn from only a Neighbour-Joining tree are not robust.
Line 180. Please, verify in the author guidelines how to cite species names.
Line 186. The prevalence information can be mentioned into the Table 1.
Line 190-230. In this section the failure in ordering the figures’ appearance in the text is particularly evident and confusing. In my opinion, the figures have a great merit, and are excellent material to describe intraspecific morphological variation to Pseudorhabdosynochus sulamericanus; however, is important show these results in the correct order. On the other hand, in this section the authors do not mention what “b” is (e.g., line 193), and do not specify the units of measurement at the beginning of this section; e.g., mean, standard deviation, maximum and minimum.
Line 263. “exceptional” is “with polymorphism morphological”? If yes, please change in order.
Discussion.
Line 302-306. The authors mentioned that: “Our phylogenetic analysis, based on COI sequences showed that H. haifensis is not closely related to H. niveatus and H. nigritus, and, in the context of available COI sequences, none of the three species is sister-species of one of the others (Figure 1), so phylogenetic similarity does not explain why they would share a putatively host-specific parasite that is not found on more closely related species.” First. Of all, as mentioned previously, the conclusions are over-drawn based on a NJ tree with no support values. Second, the “loss” of association between the non-sister taxa is not evidence against host specificity. The authors mention that host-specificity is strictly “one host species-for-one parasite species”. However, several works revise this concept to break the set “paradigm”. For example:
> Page RDM, 2003. Tangled Trees: Phylogeny, Cospeciation, and Coevolution. Chicago: University of Chicago Press, 350.
> de Vienne DM, Refrégier G, López-Villavicencio M, Tellier A, Hood ME et al., 2013. Cospeciation vs host-shift speciation: methods for testing, evidence from natural associations and relation to coevolution. New Phytol. 198:347–385.
Furthermore, at the moment the phylogeny inferred with only one gene for marine fishes means the conclusions are at best preliminary, as the authors themselves acknowledge (lines 383-384), but do not elaborate on. Perhaps it would be better if the authors do not over-extend their conclusions and explain that using only one gene fragment – and on top of that from the mitochondrion – has several drawbacks, such as inferring the gene’s (rather than the species’) evolutionary history, which in the case of the mitochondrion can display phenomena such as incomplete lineage sorting and introgression. I therefore suggest the authors tone down their conclusions on evolutionary histories and host-specificity, instead focussing more on the pattern of the apparent disjunct distribution.

Line 307-322. I find it very important that the authors mention the specificity pattern that was previously detected by Rhode 1988. As such, it is important that this idea is mentioned in the introduction as a hypotheses that can be tested based on phylogenetical analyses of the hosts (Hyporthodus spp.), to a morphological correlation with their parasite (Pseudorhabdosynochus sulamericanus).

Line 349. Please, pay attention to how you are citing the literature based on the journal guidelines. The same applies to the legend of the figure 7.

Lines 350-373. This whole paragraph comes as a surprise in the Discussion, which should not introduce new ideas, methods/results. The authors should state that they compiled the data for Table 4 since the Materials and Methods section, present the results in the Results section and only draw conclusions on these results in the Discussion.

In general, the three hypotheses that the authors mention can be plausible; however, based on the results obtained – without showing a population genealogy for Pseudorhabdosynochus sulamericanus – as it is now, the conclusions remain speculative. The authors can revise recent literature to help explain the widespread distribution of the species and host specificity to explain the speciation patterns between host-parasites association. For example:
> Martínez-Aquino, A. 2016. Phylogenetic framework for coevolutionary studies: a compass for exploring jungles of tangled trees. Current Zoology.
> Drinkwater, B., Qiao, A., and M.A. Charleston. 2016. WiSPA: A new approach for dealing with widespread parasitism. arXiv:1603.09415v1
Tables.
Table legend 1. Please delete common names of the species. Delete the column of “Date” (it is not necessary, the authors already mention the months and years of collection in the text). In the table, please modify the “Id” that corresponds to the original data obtained in this study, such as to complete the correspondence to Figure 1. For example, the authors can use only 1 capital letter plus three lowercase letters for all “Id”s.
Table legend 2. Delete the common names of the species. Change “Origin” to “Collection sites”. In the table again modify the “Id” as mentioned previously. Use italics in the table for the scientific names. Delete the “Date” column.
Table 3. Delete “Slides1999.1.6.1-3” and “HEL 460 HEL 459”.
Table 4. Please, pay attention in the style “;” throughout the table.
Figures.
Figure 1. The scientific names of species of Hyporthodus located on the right side plus the black lines are confusing because they do not correspond to clade of each taxa. Please, delete this names. On the other hand, the “Id” does not correspond with the entries in Table 1. Use italics in the figure for scientific names. Ideally this whole figure should be replaced with a ML tree and the visual presentation improved.
Please modify the “Id” that corresponds to the original data obtained in this study so that it coincides with Table 1. For example, you can use only 1 capital letter plus three lowercase letters for all “Id”s. Finally, please mention in the figure legend what the asterisk “*” stands for.
Figure 3. It would be a good idea to show the drawings of the vagina organized based on countries. In this context, it is possible to detect the high morphological polymorphism.
Legend figure 4. Change “Homologies of various” to “Similitude of various”.
Figure 5. Please, organize the composition of the figures to obtain a better idea of the comparison between the morphological traits.
Figure 6. Again, please organize each figure for a better morphological comparison, staring with the top left and ending bottom-right.
Figure 7. Please do not use italic font in the figure legend when writing “in the Atlantic Ocean and Mediterranean Sea, and localities where specimens of”, but do use italics for scientific names. In the figure legend, please change “H. niveatus + H. nigritus” to “H. niveatus and H. nigritus”.

---

## Round 0.2 · Minor Revisions

· Academic Editor

Minor Revisions

This time I have resent the authors responses to the two reviewers that had most changes proposed. Both of them now agree in that the authors had given answer to all their queries. Nonetheless, reviewer 1 raises a point that is important, although with this data set this detail may not change the results (trees) obtained, it is important that in the published literature all procedures are correctly followed and explained, so that future works based on already published ones do not repeat mistakes or incorrectly done procedures.

So, now I ask for a minor change, that will be easily resolved. For me is not so important that the model used in NJ is Kimura 2 parameters and for ML another one, since, when calculating distances a very complex model is rarely making better the estimate of distances. However, it is important that the evolutionary model used for ML is justified, either because the authors decide to use the more complex model (GTR) and so the inference algorithm estimates the parameters and hence the best model, otherwise you can use the option in MEGA6 (in menu Models, first option "find best...") that finds the best model for your data (maybe you already have done that), once found you use the best model and (very important) explain it on the material and methods. For example: "the best evolutionary model for the data set was estimated in MEGA 6 under the AIC (or the BIC) criteria.... ". Once this modification has been made the ms will be ready for publication.

Reviewer 1 ·

Basic reporting

First of all, I’d like to thank the authors for carefully taking into account the suggestions of reviewers and editor. My positive stance towards this manuscript has certainly not changed.

Experimental design

However, I feel there’s still one minor issue that needs to be resolved, regarding the phylogenetic analyses. Firstly, in this regard I’d like to say that I in a way sympathise with the authors in originally not wanting to provide a real phylogenetic analysis, but perhaps just a visualisation of barcoding data/distances. In this respect, I can even understand why they didn’t include bootstrap values etc. – but then I think that some aspects of the tree, like including an outgroup and mentioning sister-group relations, gave the impression that at least part of the intention was phylogenetic analysis, too, and then of course somewhat more thorough analyses are needed. Anyways, this seems to be solved now. Secondly, one important comment that has not been taken into account, is the question how the various models of molecular evolution were chosen. This raises question, even more so in the revised manuscript, since the authors apply a different model for the ML and NJ analyses. Since they are using the same software on the same dataset, it seems strange to me that two different models were chosen as the optimal one(s).

Validity of the findings

Otherwise, no further comments.

Additional comments

Otherwise, no further comments. Manuscript still merits publication but please solve the issue of the optimal model of molecular evolution.

Reviewer 3 ·

Basic reporting

Complete and satisfactory

Experimental design

Complete and satisfactory

Validity of the findings

Complete and satisfactory

Additional comments

The authors responded to the revisions in a satisfactory manner.

---

## Round 0.3 · accepted · Accept

· Academic Editor

Accept

Thanks for the fast answers with the changes required. The ms is now accepted.